# Phenotypic Diversity of Morphological Characteristics of Pitahaya (*Selenicereus Megalanthus* Haw.) Germplasm in Colombia

**DOI:** 10.3390/plants10112255

**Published:** 2021-10-22

**Authors:** Ana Cruz Morillo-Coronado, Elsa Helena Manjarres Hernández, Lucimar Forero-Mancipe

**Affiliations:** Grupo CIDE Competitividad Innovación y Desarrollo Empresarial, Universidad Pedagógica y Tecnológica de Colombia, Tunja 150003, Colombia; elsa.manjarres@uptc.edu.co (E.H.M.H.); lucimar.forero@uptc.edu.co (L.F.-M.)

**Keywords:** germplasm, *Selenicereus megalanthus*, morphoagronomic descriptors, phenotypic variation, genetic diversity

## Abstract

Yellow pitahaya is an exotic fruit that is rich in essential nutrients and antioxidants. In Colombia, it grows naturally in warm areas, but it is not clear which species exist because this genus presents a lot of intra and interspecific hybridization. More studies are needed in this field along with characterizations of the genotypes. This study aimed to undertake an in situ morphoagronomic evaluation of yellow pitahaya genotypes from five municipalities in Boyacá, Colombia. Measurements were taken in a completely random design. Qualitative and quantitative descriptors for cladodes, fruits and covered production systems were evaluated with a descriptive analysis, Spearman correlation variance, and multivariate and cluster analysis. The results showed that cladode characteristics such as cladode width, distance between areoles, number of spines, length of areoles, margin ribs of cladode and waxiness could be used to identify *Megalanthus* spp. Characteristics such as weight (270–274 g), size (100 mm), color of the fruit and pulp as well as acidity (0.18) and soluble solids (15.7) were highly variable between the genotypes. Genotypes with good morphological and fruit characteristics were identified (Gen2 and Gen9), which can provide the bases for the selection of pitahaya that satisfy the needs of farmers and consumers.

## 1. Introduction

The Dragon fruit, known as pitaya or pitahaya, belongs to the Cactaceae family, which originated from the southern and central regions of Mexico and America, and is separated in two genera: *Hylocereus* and *Selenicereus.* It is a nutritious and exotic fruit cultivated throughout the tropical and subtropical regions of the world. Pitahaya production has attracted interest in the United States, Australia, Southeast Asia, Israel and other regions [1]. This fruit has gained considerable attention from consumers because it is a unique fruit that can tolerate drought stress and contains a considerable amount of nutrients. It is rich in polyphenols, vitamins, sugar, amino acids and betalain pigments [2]. Furthermore, pitahaya fruit contains substantial amounts of unsaturated fatty acids (linoleic and linolenic) with broad applications in therapeutic and cosmetics preparations [3]. It has drawn worldwide attention because of its new flavor, color and attractive appearance, along with enormous health benefits [4]. It gained commercial potential in different countries as the result of consumer preference for new, exotic, and phytochemically rich fruits and its adaptability to new environments with abiotic stress tolerance, such as droughts and extreme temperatures [5].

Generally, differences in pitahaya germplasm can be easily shown with some distinguishing phenotypic characteristics, such as fruit size, fruit color, and number of spines at the areola that formed on branch/stem segments [6]. Nowadays, it is very difficult to separate species and varieties of the dragon fruit because of high intra and interspecific hybridization that has created some taxonomical confusion worldwide [7]. Morphological and genetic heterogeneity in many fruit characteristics, such as sweetness, size, shape, color, and bracts number, resulting from intra and inter-specific hybridization that makes it difficult to increase quality standards for the exportation market, posing serious problems when determining performance in handling and shelf-life [8].

Internationally, a large collection of pitahaya germplasm accessions is curated at the University of California South Coast Research and Extension Center (SCREC), Irvine, CA. The collection was first established in 2005 and includes seven varieties native to Nicaragua, with approximately 120 individual plants representing four different species (*H. undatus*, *H. polyrhizus*, *H. costaricensis*, and *Hylocereus* sp. Unnamed), two varieties native to Mexico with 34 individual plants representing two species (*H. ocamponis* and *H. megalanthus*), two varieties from San Diego with 34 individual plants representing two species (*H. undatus* and *H. guatemalensis*), and seven varieties from Florida, including 120 individual plants representing four species and hybrids (*H. undatus*, *H. guatemalensis*, *H. megalanthus*, and several putative hybrids identified as *Hylocereus* spp.). Additional accessions have been added intermittently to the collection, including 17 additional accessions from Nicaragua (within the *H. polyrhizus*/*costaricensis* group) [9]. These accessions were tentatively labeled *H. costaricensis*/*polyrhizus* because both species are commonly found in Nicaragua and throughout Central America. An additional variety was purchased from Mexico (17 individual plants representing *H. ocamponis*) and several other varieties were also sourced from Florida. Additional material has been obtained from local producers, and two plants per accession have been added to the collection. In total, the collection now includes 378 individual plants, potentially representing 54 varieties and seven different species [10].

Conventionally, morphological traits had been used to differentiate plant germplasm/species and to elucidate their genetic relationship [11]. The great morphological diversity between *Hylocereus* and *Selenicereus* species has been reported at the intraspecies and intravarietal levels, as a result of their coevolutionary process with the environment, which makes the production of new varieties extremely difficult. This led the International Union for the Protection of New Varieties of Plants [12] to develop a guide to document how new varieties of pitahaya are determined. However, as mentioned above, intraspecies/intravarietal morphological differences between vegetative clones and hybridization within this group lead to genetic mosaics between new lines and make identification between varieties extremely difficult.

Mainly because pitahaya species have easily hybridized since the late 1980s and early 1990s, breeders in the United States, Israel, and Southeast Asia have developed several hybrids [13], including crosses between either *H. guatamalensis* or *H. megalanthus* with *H. undatus* as the other parent, which have resulted in plants with great adaptability and high fruit quality [9].

In Colombia, the germplasm bank of yellow pitahaya and its wild relatives is located in the facilities of the National University of Colombia, Palmira. It has 300 introductions, both cultivated and wild, of yellow pitahaya (*S. megalanthus* 238 accessions) and red pitahaya (*H. undatus*, *H. costaricensis*, *Hylocereus* spp. 36 accessions). All of these are properly coded; however, not all of them are characterized [14].

In this country, knowledge of this crop mainly comes from the empirical processes of farmers who are motivated by the prices that fruits can fetch in some of the months of the year; however, increased penetration into international markets requires research on the processes of propagation, obtaining elite material, and resistance to biotic and abiotic factors, among other topics [15]. However, one of the bigger limitations is the broad morphological variation seen in the vegetative structures, which leads to confusion in identifying each species, with a lack of consensus [16], where classification is mainly based on the number of areola ribs, the contour of the stem, the relative firmness of the stem and the size and color of the fruits; in addition, various studies on domesticated cactus species have demonstrated variations in fruit characteristics related to the domestication process, resulting in a lack of a taxonomic database [1]. 

Studies on morphological characterization of germplasm worldwide have shown great variation in the evaluated accessions that can be conserved and used in the genetic improvement of the species; for example, a study evaluated morphological, biochemical and molecular characterizations of four dragon fruit (*Hylocereus* spp.) genotypes grown in Andaman and on Nicobar Island and revealed the presence of a considerable amount of genetic variations that could be used as key traits for distinguishing three different species [8]. Four dragon fruit genotypes: *H. polyrhizus*, *H. megalanthus* and two *H.* hybrids resulted in different plant growth and development. *H. polyrhizus* had the best plant growth; whereas, *H. megalanthus* had the lowest plant growth. Therefore, the red variety was more suitable for cultivation in Pangandaran [17]. The differences in the anatomical structure and morphology of the plants could cause differences in its optimal growth location. Morphological and agronomic characteristics can be used to determinate genetic variation in a single population.

Despite the productive potential, a limiting factor in the development of this crop in Colombia is the incidence of pest and diseases, low fruit quality, the technological level, the associativity and the lack of cultivated material, generating significant losses in yield [18,19]. Different research institutions and universities have tried to find a solution to these problems [14,18]. These studies have shown that there is no certified planting material and that only a few are grown by farmers, generating vulnerability to different phytosanitary problems, for which it is necessary to carry out genetic studies that lead to the identification of elite materials that meet the needs of the productive chain of yellow dragon fruit in Colombia since the genetic base of germplasm resources is limited.

A study on genetic diversity was carried out [19] in yellow dragon fruits using morphological descriptors, showing the existence of phenotypic variability in the evaluated materials. However, in Colombia and especially in Boyacá, which is one of the main producing departments, genetic studies on this species are scarce; there are more studies on red pitahaya in the Department of Antioquia [20].

In this context, this research evaluated the genetic diversity of yellow dragon fruit materials from different municipalities in the Department of Boyacá, Colombia, with morphological descriptors to understand the genetic background of this germplasm and provide theoretical orientation for the planning of conservation strategies, use and plant breeding.

## 2. Results

The climatic conditions of the study region were characterized by few fluctuations in the photoperiod and average temperatures. The minimum temperature during this study ranged between 13 and 16 °C, and the maximum was between 19 and 26 °C, with an average temperature of 20 °C. The monthly precipitation was less than 1 mm. The average relative humidity was 78%, and the daily illumination was 5.3 h a day. In general, the monthly climatic data in these variables did not show large fluctuations, but remained at the expected values for the study area during the evaluated time period.

### 2.1. Morphoagronomic Characterization Using Quantitative Descriptors

The quantitative morphoagronomic and fruit descriptors evaluated in the yellow pitahaya genotypes showed a broad range of variation. The variables that showed a high coefficient of variation were bract color index (BCI) and number of fruits (NF) with values of 68.9% and 51.5%, respectively (Table 1). These variables are mainly associated with the fruits. The genotypes with the highest number of fruits per cladode were 6, 5 and 10, respectively. The genotypes with the best averages of fruit weights were genotype 2 with 274.10 g per fruit and genotype 9 with 270.10 g per fruit; while the genotypes with the lowest fruit weight were genotype 3 with 161.50 g per fruit and genotype 6 with 180.80 g per fruit. The lowest degree of acidity, an important descriptor for the commercialization of the fruit, was in genotypes 2, 4, 5 and 6, with a value of 0.18% (Table 1). 

Spearman correlation analysis (*p* ≤ 0.05) between the morphoagronomic quantitative variables showed that there were positive, high and significant correlations between leaf rib width (LRW) and number of spines per areola (NSA) (r = 0.81), number of fruits (NF) and number of spines per areola (NSA) (r = 0.72) (Figure 1a). In the fruits, there were high and significant correlations between equatorial bract length (EBL) and equatorial bract width (EBE) (r = 0.84), fruit weight (FWE) and shell weight (SW) (r = 0.8), fruit weight (FWE) and fruit width (FW) (r = 0.77), fruit weight (FWE) and pulp weight (PWF) (r = 0.76). Furthermore, in the fruit variables, there were negative and significant correlations between fruit length/width ratio (FRLW) and fruit width (FW) (r = −0.85) and bract color index (BCI) and fruit width (FW) (r = −0.62) (Figure 1b).

Principal component analysis showed that 52.8% of the total variance was explained in the first two components (CP1 = 29.1% and CP2 = 23.7%) (Figure 2a). The variables that made the greatest contribution to the variation of CP1 were the fruit variables, such as pulp weight (PWF), fruit width (FW), equatorial bract width (EBE), fruit length (FL), equatorial bract length (EBL) and fruit weight (FWE). CP2 saw both fruit and morphoagronomic variables, such as number of bracts (NB), height undulations between successive areoles in a rib (HUA), distance between areoles (DBA), longest spine length (LSL) and leaf rib width (LRW).

The cluster analysis of the quantitative fruit and morphoagronomic variables grouped the genotypes into five clusters (Figure 2b). However, the groups were not established according to the collection area or place of origin but rather by the main morphological characteristics of the evaluated genotypes. The first group had genotype 2, which presented the best characteristics with respect to the fruits with average weights of 274.10 g, pulp weight (PWF) of 148.83 g, fruit length (FL) and fruit width (FW) of 100.29 and 70.15 mm. The second group was represented by genotype 9, which had the highest soluble solids (SS) with respect to the other genotypes with 17.03° brix, weight of the fruit pulp (PWF) with 156.33 g, and fruit weight (FWE), 270.10 g. The genotypes in the third group were 5, 6, 3 and 11, which presented an average fruit width and length of 93.02 and 55.81 mm, fruit weight (FWE) between 161.50 and 182.60 g, average number of fruits (NF) of 3 per cladode, and soluble solids (SS) between 12.43 and 16.97° brix. The genotypes of the fourth cluster were 1, 4 and 8, which were characterized by a fruit weight (FWE) between 193.75 and 206.20 g, average acidity (A) of 0.22%, pulp weight (PW) between 105.67 and 129.17 g, average length of the cladodes of 95.63 cm, and average pericarp thickness of 3.56. Finally, group five had the genotypes 7 and 9, which presented an average length of cladode of 123.52 cm, distance between areolas (DBA) of 4.77 cm, length and width of the fruit average 98.76 and 62.90 cm, respectively, and pulp weight (PWF) between 129.00 and 156.33 g. These analyses were consistent with the principal component analysis since the variables that contributed the most to the observed phenotypic variation were also important in determining the clusters.

### 2.2. Morphoagronomic Characterization Using Qualitative Descriptors

The evaluation of the qualitative variables showed that the most common fruit shape was elongated, while genotypes 2 and 9 had a round fruit shape (Table 2). In the evaluated genotypes, there were no waxes in the cladodes. The most variable qualitative descriptor was the color of the spines. The genotypes of which the individuals showed a stable spine color were genotypes 1 and 6 with 100% light brown color, and genotype 2 with (100%) dark brown color. The other genotypes presented variations in the evaluated individuals. In the texture of the surface of the cladodes, the genotypes presented 74.5% smooth surface, while genotype 6 presented a rough surface.

The multiple correspondence analysis showed that 54.7% of the total variance observed was explained in the first two components CP1 (33.5%) and CP2 (21.2%). The first component grouped the genotypes according to characteristics such as rough surface texture of the cladode (STF), absence of pigmentation at the tips and margins of vegetative shoots (PTM) and gray areola coloring (AC). The second component had intense pigmentation at the tips and margins of the vegetative shoots (PTM) and an opaque brown color of the thorns (TC). Figure 3a shows the distribution of the variables according to their contribution to the total variance in the first two components. 

The cluster analysis of the qualitative variables formed seven groups (Figure 3b). The first one had genotype 6, which was characterized by an elongated fruit shape, light gray areola color and light brown spine color. The second group had genotype 3, which presented 90% of the fruits with an elongated shape, 70% of the individuals with a concave shape of the margin between areoles (SMA), and 80% of the color of the dark gray areolas. The third group had genotype 2, which was characterized by a round fruit shape, 100% smooth surface texture of the cladode, dark gray areoles, and dark brown spines. The fourth group was represented by genotypes 4 and 11, which were characterized by dull brown, light brown and dark brown spine colors, and 85% elongated fruit shape. In the fifth, genotypes 8 and 9 grouped with dark gray areolas and smooth surface on the cladodes. Group six was represented by genotype 7, which was characterized by an elongated fruit shape and smooth surface texture on the cladodes. Group seven had genotypes 1, 5 and 10, which presented an elongated fruit shape and smooth surface texture on the cladodes.

### 2.3. Morphoagronomic Characterization Taking into Account the Joint Analysis of Qualitative and Quantitative Descriptors

The factorial analysis of mixed data considered all quantitative and qualitative descriptors, differentiating the genotypes with outstanding morphoagronomic and fruit characteristics. This analysis showed that the contribution of the variables to the first two components was 29.9%. The descriptors that contributed positively to CP1 (16.7%) were fruit width (FW), soluble solids (SS) (quantitative variables), smooth surface texture of the cladodes (STF), and elongated fruit shape (FS) (qualitative variables) (Figure 4a). For CP2 (13.2%), the quantitative variables included the peel/pulp ratio (PPR), number of bracts (NB) and acidity (A), and the qualitative variables were shape of the margin between areolas (SMA) and pigmentation at the tips and margins of vegetative shoots (PW). 

The cluster analysis formed six groups (Figure 4b). The first group had genotype 11, which presented a high variability in the qualitative descriptors, fruit weight lower than the average (169.40 g), fruit acidity level (A) of 0.23%, and the lowest weight in the fruit peel (43.33 g). The second group had genotype 7, which had a fruit weight of 199.17 g, pulp weight higher than the average (129.00 g), as well as length of the cladode (127.05 cm), elongated fruit shape, and smooth surface on the cladodes. The third group was represented by genotype 2 with a fruit weight of 274.10 g, pulp weight (PWF) of 148.83 g, fruit length (FL) of 100.29, light brown spines, round fruit shape, smooth cladodes, and dark gray areolas. The genotypes that made up the fourth group were 8, 4 and 9, which had dark gray areoles, smooth surface texture, fruit weight between 193.75 and 270.10 g, and average fruit pulp weight of 124.33 g. The fifth group had genotype 6 with an elongated fruit shape, fruit weight of 180.80 g, light gray areola color, and light brown spines color. The sixth group had genotypes 1, 3, 5 and 10, which had an average cladode length of 110.99 cm, fruit weight between 161.50 to 215.60 g, average fruit acidity of 0.21%, soluble solids of 16.22° brix, an elongated fruit shape, and smooth cladodes.

## 3. Discussion

Yellow pitahaya is a popular crop in Colombia because of its nutritional value and productive potential. Recently, farmers have begun growing different genotypes in the producer municipalities. The morphological characteristics of these genotypes have yet to be elucidated [19]. Colombia has developed vastly with the cultivation of several dragon fruit genotypes but the morphological characteristics and adaptation to different altitudes have not been clearly studied [18]. Morphological and agronomic characteristics can be used to measure genetic diversity in a particular individual population. The plant phenotype is a form of plant adaptation to environmental conditions [17].

This study analyzed the characteristics of 11 yellow pitahaya genotypes in five municipalities in the Department of Boyacá to find out which one is optimal since one of the main problems of the study area is that there is no such variety. The planting material was introduced to the region, perhaps through the exchange of seeds between producers in the country. It was stated by [19] that there were several morphological characteristic that can be described to distinguish different types of dragon fruit species. Eleven genotypes of yellow pitahaya have shown different plant characteristic. It was stated by [21] that the main differences of *Hylocereus* species were the size and color of the fruits and also the shape and number of spines. This statement corresponded with the current analysis of eleven *Selenicereus* genotypes, in which the differences in the stem shape and number of spines were observed. Similar research was conducted by [20,22], who stated that the number of spines was a reliable characteristic to describe *Hylocereus*. The four dragon fruits grown in Pangandaran have similarities in the number of spines per areola bur differ in the shape and color of the spines [17].

In the case of qualitative traits, most of the genotypes exhibited an elongated fruit shape (Gen1, Gen5, Gen6 and Gen7), except Gen2 and Gen9 with a round fruit shape (Table 2). None of the genotypes had wax but there was variation for each of the categories of the morphological descriptors evaluated. It was stated by [8] that cladode, floral and fruit characteristics of *H. megalanthus,* such as margin ribs of cladode, waxiness, sepal color, color of ring at base of reproductive organs in flower, fruit shape, position towards peel, pulp color, peel color and seed size are visible taxonomic traits to distinguish this species from two *Hylocereus* spp.: *H.undatus* and *H. costariscensis*.

Cladode characteristics such as cladode width (mm), distance between areoles (mm), number of spines, length of areoles (mm), margin ribs of cladode and waxiness could be used to identify *Hylocereus* spp. and *Megalanthus* spp. [8,17]. In this study, the genotypes showed the shape of margin between areolas between concave and right, contrary to that reported in other studies on different species of *Hylocereus*, where the predominant forms were convex, with the presence of wax [8]. The cladode width and distance between areoles corresponded well with earlier studies from México [23], India [8], Puerto Rico and Colombia [6,18,19,20].

It was stated by [24] that pitahaya is a dispersed crop species in the tropics and subtropics that presents high polymorphism. The species has undergone human selection through the action of collecting fruits, which promoted the diversity of fruits in shape, size, color and organoleptic quality. Today, more than one species of pitahaya is known. Morphological and genetic heterogeneity in many fruits characteristics such as sweetness, size, shape, color, and bracts number by this intra and inter-specific hybridization [25] makes it difficult to increase the quality standards for the exportation market because it poses serious problems in determining their performance in handling and shelf-life. Fruit morphology, such as size, color of peel and pulp and fruit color are the main taxonomic evidence to differentiate several *Hylocereus* spp. and exhibit the external quality of fruits [26].

The presence of a high number of natural pollinators such as honey bees in the field plays a major role in fruit set in pitahaya fruits [27]. Total soluble solids, being the most desirable characteristic for consumer preference, is measured as Brix, which can be affected by a set of factors, such as genetic, climatic, soil, and management, among others [6]. In the present study, the SS ranged between 12.43% and 16.97%, representing better fruit quality that evidenced the earlier report that SS values between 11 and 15% have good market preference [18].

For fruit weight, most of the genotypes had values above 100 g, a desirable characteristic for the market. Individuals 2 and 9 presented the best characteristics for the market, such as fruit weight, pulp weight, fruit diameter and a high content of soluble solids, which would be of interest for distribution. 

The multivariate and cluster analyses showed that the cladode and fruit characteristics showed higher variability among the morphological traits (Figure 5a,b). Although many authors find positive results, morphological and agronomic characteristics used to measure genetic diversity in certain populations of individuals often do not allow identification of discrete taxonomic groups since most plant characteristics are influenced by environmental factors, exhibiting continuous variation and a high degree of phenotypic plasticity [28]. This study demonstrated that the natural populations of pitahaya from the five departments showed high variation in the morphological characteristics that were evaluated in situ. Although the five studied sites have similar climatic conditions, the same variables should be evaluated in accessions established in a single environment, thereby confirming whether the variation might have been affected by uncontrolled environmental factors in this observational study, such as different soil types [29].

Studies in other countries have found high variation in characteristics of agronomic importance, even within the same species of *Hylocereus* spp. and *Megalanthus* spp. [6,8,20], which is favorable for future breeding studies. The availability of this information would be of great assistance in developing an appropriate method for the cultivation of a particular species [29,30]. Several studies related to the diversity of dragon fruit have been reported. A study was reported by [31] on variations in dragon fruits based on morphology, isozyme, and vitamin C content in the area of Pasuruan (East Java), Sukoharjo, Klaten (Central Java), and Bantul sub-districts (Yogyakarta). A study was reported by [26] on pollination methods on fruit set and fruit characteristics in several Pitaya clones, which aimed to improve pollination efficiency, fruit quality, and yield by determining pollination agro-management requirements. The dragon fruit plants planted in Pasuruan, Sukaharjo and Bantulhave had significant differences in stem morphology between the different species/varieties [10]. The variations in stem morphology such as curvature of the stem, margin hardness (presence of sclerenchyma), distance between areoles, number of spines, rib height, rib thickness, length, and color of the stem are important for species differentiation [20].

Apart from differences in species or accessions, differences in fruit morphology can be related to changes in the physiological level of dragon fruit at various stages of fruit development [31]. The main differences among several *Hylocereus* species were the size and color of the fruits and the number and form of the spines [21]. Also, the variety and flowering time have a large influence on the physio-morphological traits of dragon fruits [4,32].

In Colombia, the morphoagronomic characterization studies of these two species have obtained results similar to those reported in this study, highlighting the existence of genetic variability that can be used in conservation and genetic improvement programs that lead to the identification of elite materials, where genotypes 2 and 9 could be a good alternative. However, it is necessary to complement these morphological characterization studies with biochemical and molecular data that better discriminate the germplasm given the limitations of this type of descriptor. Research on yellow pitahaya should be intensified and extended by emphasizing its value chain and production aspects for a long-term perspective.

## 4. Materials and Methods

For the morphological characterization in situ, the sampling of the yellow pitahaya was carried out in the main producing municipalities in the Department of Boyacá (Zetaquira, Páez, San Eduardo, Paéz and Miraflores). In total, four municipalities and 22 farms were sampled (Table 3 and Figure 6), with 11 genotypes belonging to *H. megalanthus*. Ten plants per farm were used for the morphological characterization during 2020–2021. The description was performed in situ. At each farm, observation tours were carried out along with consultations with the local people in the Pitafcol association (Yellow Dragon Fruit Producers Association) about the location of farms. Individuals were randomly selected at each farm, were georeferenced using global positioning equipment (GPS Garmin^®^), and were described. Additionally, the passport data of the evaluated individual was produced.

Qualitative and quantitative descriptors were evaluated for cladodes, fruits, and covered production systems. In total, seven characteristics were recorded in the characterization of the cladodes: one quantitative and six qualitative (Table 4). Measurements were taken in a completely random design, selecting 10 plants from each farm and production system. The experimental unit was each selected plant. Six fruits were taken from each farm for the morphological characterization, which had the same maturation stage, that is to say the entire fruit had an intense yellow color that is characteristic of the *Selenicereus megalanthus* species, and the descriptors in Table 2 were measured.

### Statistical Analysis

The statistical analyses carried out for the first data from the morphological characterization were used in the descriptive analysis for the qualitative and quantitative descriptors in InfoStat. The assumptions for the parametric analyses were verified, along with the analysis of variance (ANOVA) [33]. For the quantitative variables, Spearman′s correlation was estimated for the quantitative variables, and its significance was assessed using the *t*-test (null hyphothesis is H0: r = 0. with 5% of significance) to identify the characteristics most associated with performance. The principal component analysis (PCA) was based on the correlation matrix between the characteristics, which were graphed on a two-dimensional plane to group the farms with R Core Team (2020). For the qualitative descriptors, a multiple correspondence analysis (MCA) and correlation analysis were carried out. The matrix distance was performed using the Euclidean distance for the cluster analysis. All analyses were carried out with the algorithms in the extra-factor package of the R program, version 1.07, for the cluster [34]. For the joint analysis of the quantitative and qualitative variables, a factorial analysis of mixed data was carried out with the factoextra package in the R program, which provided functions to extract and visualize the output of multivariate data analysis, including PCA, CA, MCA and FAMD (factor analysis of mixed data). The distance matrix was performed using the Euclidean distance for the cluster analysis. The minimum Ward´s distance was used as the grouping method, where each cluster was given by the smallest increase in the total sum value of the squares of existing differences within each cluster of each observation for the cluster centroid. Additionally, a dendrogram was generated using the Euclidean distance and hierarchical grouping method of Ward´s minimum variance with Facto Mine R package, which was used when the variables were grouped together [35]. 

## 5. Conclusions

The descriptors, such as a fruit weight, fruit shape, pulp weight, surface texture of cladode, spine color and areola color differentiated the yellow pitahaya genotypes. The broad phenotypic variability in the genotypes must be exploited to create strategies for finding solutions for the principal limitations of the crop and to meet the needs of farmers, producers and consumers. In addition, the genotypes (Gen 2 and Gen 9) that were identified can be a productive alternative for the generation of new and better varieties.

## Figures and Tables

**Figure 1 plants-10-02255-f001:**
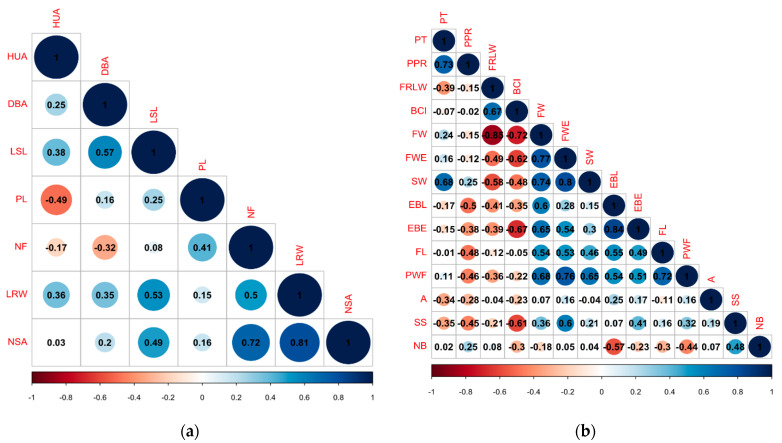
Spearman’s correlation between quantitative variables. (**a**) Morphoagronomic variables. (**b**) Fruit variables.

**Figure 2 plants-10-02255-f002:**
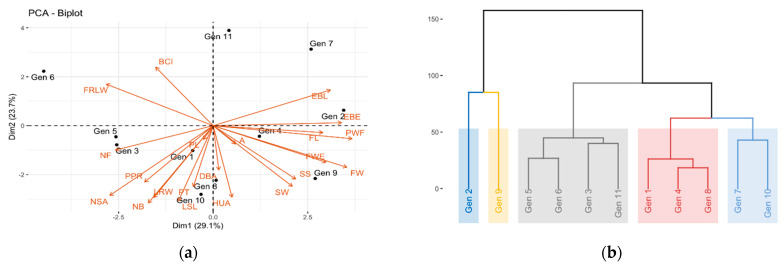
(**a**) Principal component analysis biplot. (**b**) Hierarchical cluster analysis of yellow pitahaya fruit genotypes.

**Figure 3 plants-10-02255-f003:**
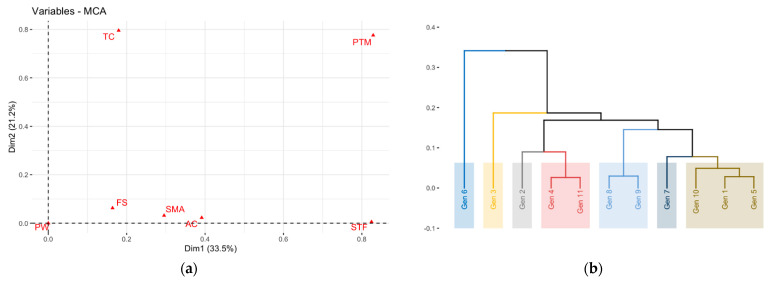
(**a**) Multiple correspondence analysis shows the contribution of qualitative variables. (**b**) Cluster analysis showing seven groups of yellow pitahaya materials formed according to qualitative variables.

**Figure 4 plants-10-02255-f004:**
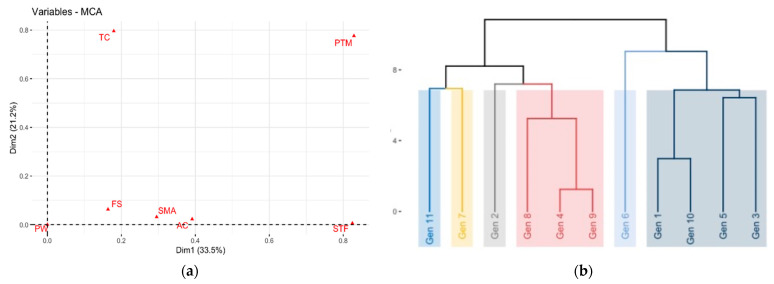
(**a**) Mixed factor analysis shows the contribution of the variables, ordering the materials according to the qualitative and quantitative variables. (**b**) Cluster analysis. showing six groups of materials formed according to qualitative and quantitative variables.

**Figure 5 plants-10-02255-f005:**
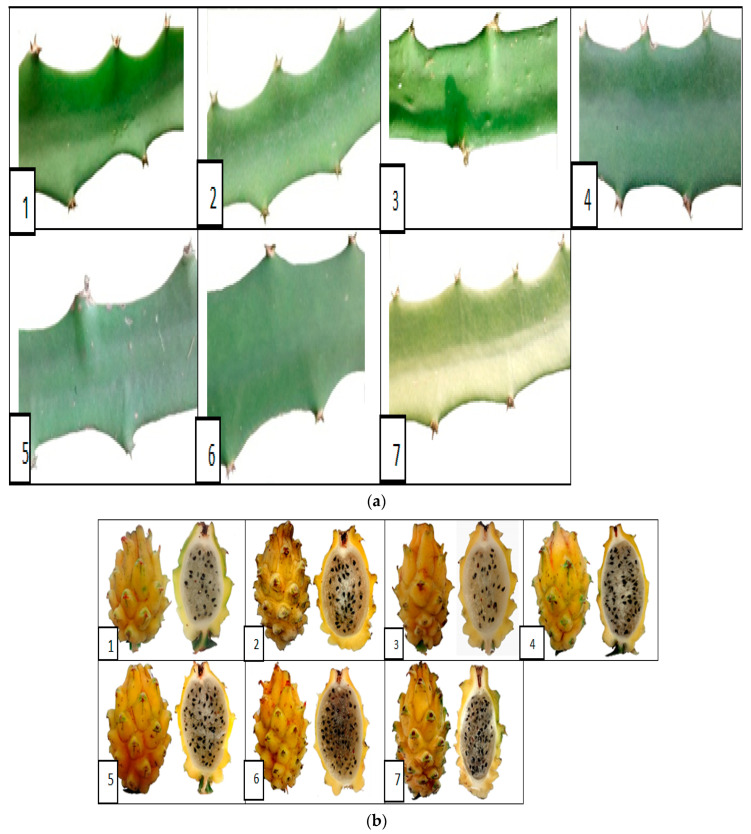
(**a**) Variation in width of ribs of pitahaya filocladodes characterized by farms: 1, 3, 4, 5, and 6 in the municipality of Miraflores, Russian village 2 municipality of Berbeo village Batatal; 7 municipality of Paéz, Yamuntá village. (**b**) Variation in pitahaya fruits characterized by farms: (1), (3), (4), (5), and (6) in the municipality of Miraflores vereda Rusa (2) municipality of Berbeo, Batatal district; (7) municipality of Paéz, Yamuntá village.

**Figure 6 plants-10-02255-f006:**
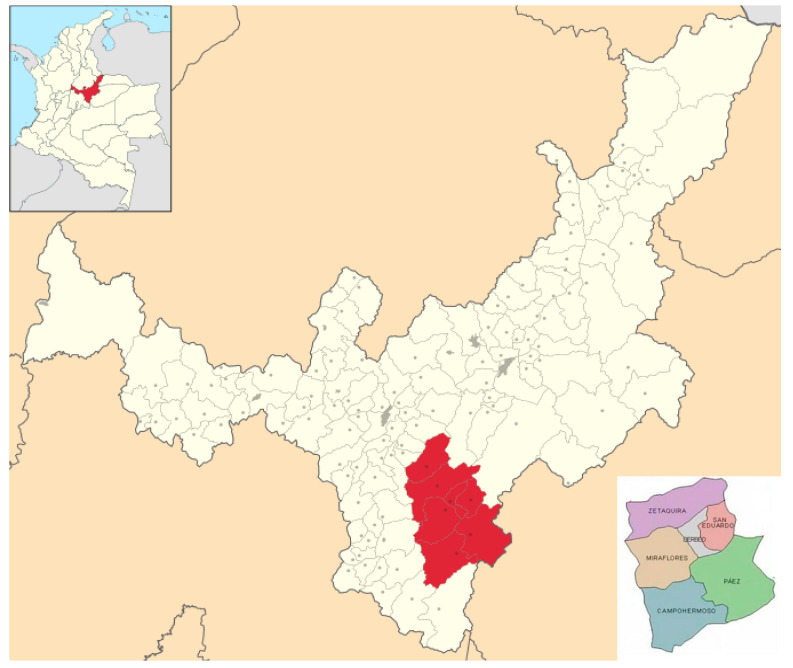
Geographic location of the province of Lengupá (Boyacá) Colombia and its municipalities.

**Table 1 plants-10-02255-t001:** Descriptive statistics of the quantitative morphoagronomic and fruit descriptors of the yellow pitahaya materials.

Descriptors	Gen 1	Gen 2	Gen 3	Gen 4	Gen 5	Gen 6	Gen 7	Gen 8	Gen 9	Gen 10	Gen 11	Average	C.V
Morphoagronomic Descriptor													
Number of fruits (NF)	2.4	1.9	2.1	2.9	3.6	4.0	1.5	3.3	2.9	3.4	2.9	3.48	51.5
Cladode length (PL)	93.28	78.52	87.34	96.64	110.22	116.22	127.05	96.97	120.87	153.1	100.95	118.11	31.4
Distance between areolas (DBA)	4.91	4.34	4.85	4.57	4.16	4.47	4.79	4.22	4.76	5.60	4.18	5.55	17.3
Leaf rib width (LRW)	5.60	4.24	4.71	5.23	5.28	4.74	4.31	5.29	4.53	5.76	4.22	5.90	27.6
Height of undulations between successive areoles in a rib (HUA)	0.29	0.17	0.33	0.31	0.16	0.10	0.15	0.36	0.27	0.23	0.17	0.4	46.8
Number of spines per areola (NSA)	3.0	2.6	2.9	2.9	3.0	3.0	2.5	2.9	2.9	3.0	2.6	2.99	10.8
Longest spine length (LSL)	0.32	0.25	0.31	0.28	0.30	0.28	0.28	0.30	0.33	0.31	0.26	0.32	21.0
**Morphoagronomic and physicochemical descriptor of the fruits**													
Fruit length (FL)	101.22	100.29	89.16	100.33	94.42	93.39	99.30	94.34	98.22	98.52	95.10	96.75	9.2
Fruit width (FW)	60.30	70.15	56.32	65.33	57.60	49.08	58.37	65.00	67.44	61.84	60.25	61.66	14.5
Fruit length/width ratio (FRLW)	1.69	1.43	1.60	1.56	1.64	2.69	1.71	1.46	1.45	1.60	1.59	1.62	35.5
Fruit weight (FWE)	204.90	274.10	161.50	193.75	182.60	180.80	199.17	206.20	270.10	215.60	169.40	207	28.9
Number of bracts (NB)	44.40	39.40	46.00	41.88	50.70	40.60	39.83	48.60	43.30	47.30	37.90	43.97	13.8
Equatorial bract length (EBL)	19.00	21.71	18.82	26.34	18.32	16.85	27.15	19.72	21.53	19.25	22.43	20.87	29.9
Equatorial bract width (EBE)	15.86	19.95	15.64	22.90	16.48	15.85	24.19	20.65	22.70	18.27	18.12	19.41	36.9
Length of longest bract (LLB)	42.23	47.05	42.92	46.83	43.28	45.38	48.35	47.15	46.10	46.71	46.93	45.61	11.4
Pericarp thickness (PT)	3.54	4.18	4.27	3.91	3.18	3.69	2.37	3.23	3.71	3.61	2.05	3.42	25.6
Shell weight (SW)	76.17	105.83	67.83	79.67	57.00	66.00	59.50	75.00	98.17	84.33	43.33	74.27	33.6
Pulp weight (PWF)	129.17	148.83	86.50	111.00	84.17	87.33	129.00	105.67	156.33	119.17	111.00	116.62	31.9
Peel/pulp ratio (PPR)	0.59	0.73	0.88	0.71	0.70	0.75	0.46	0.73	0.63	0.67	0.39	0.65	32.2
Soluble solids (SS)	15.80	15.73	15.29	15.67	16.97	12.43	15.85	16.30	17.03	16.83	15.70	15.94	12.4
Acidity (A)	0.20	0.18	0.21	0.18	0.18	0.18	0.24	0.29	0.19	0.25	0.23	0.21	24.4
Bract color index (BCI)	1.58	1.05	1.28	1.00	1.18	1.37	1.27	0.37	0.67	0.86	1.27	1.07	68.9

CV Coefficient of variation.

**Table 2 plants-10-02255-t002:** Qualitative descriptors used in the morphoagronomic characterization of the yellow pitahaya genotypes in the Department of Boyacá, Colombia.

Descriptors	Category	Gen 1	Gen 2	Gen 3	Gen 4	Gen 5	Gen 6	Gen 7	Gen 8	Gen 9	Gen 10	Gen 11
Fruit shape (FS)	Elongated	100	---	90	70	100	100	100	40	---	60	80
	Round	---	100	10	30	---	---	---	60	100	40	20
Surface texture of the cladodes (STF)	Smooth	100	100	50	100	50	20	100	90	100	100	90
	Rough	---	---	50	---	50	80	---	10	---	---	10
Presence of wax (PW)	Absence	100	100	100	100	100	100	100	100	100	100	100
Shape of the margin between areolas (SMA)	Concave	80	40	70	80	20	20	30	90	80	60	90
	Convex	---	---	---	---	---	---	---	---	---	---	10
	Right	20	60	30	20	80	80	70	10	20	40	---
Areola coloring (AC)	Light grey	100	---	20	10	50	100	30	---	---	70	60
	Dark grey	---	100	80	90	50	---	70	100	100	30	40
Thorns color (TC)	Dull brown	---	---	40	20	10	---	---	10	10	---	10
	Bone brown	---	---	10	---	---	---	10	---	---	---	---
	Light brown	100	---	---	10	60	100	60	50	80	70	30
	Brown	---	---	40	---	---	---	---	30	10	---	---
	Dark brown	---	100	10	70	30	---	30	10	---	30	60
Pigmentation at the tips and margins of vegetative shoots (PTM)	Absence	20	40	10	30	---	60	---	---	---	---	---
	Light	60	60	40	40	50	20	40	10	100	50	50
	Intense	20	---	50	30	50	20	60	90	---	50	50

The values indicate the percentage of introductions that present the respective category.

**Table 3 plants-10-02255-t003:** Origin sites of the evaluated of yellow pitahaya in the Department of Boyacá.

Origin	Quantity	Height	North Latitude	West Longitude
Miraflores (Gen1. Gen2. Gen3. Gen4. Gen6. Gen7)	6	1432	5°11′47″	73°8′40″
Zetaquira (Gen 5)	1	1665	5°17′2″	73°10′13″
Páez (Gen8. Gen9)	2	1335	5°13′36″	73°7′36″
Berbeo (Gen 10. Gen11)	2	1300	5°5′56″	73°3′6″

**Table 4 plants-10-02255-t004:** Morphoagronomic descriptors used for the characterization of yellow pitahaya materials and morphoagronomic and physicochemical descriptors of the fruits of the yellow pitahaya materials.

Quantitative	Acronyms	Unit of Measurement	Qualitative	Acronyms
Morphoagronomic Descriptors				
Number of fruits	(NF)	#	Surface texture of the filocladiolus	(STF)
Philocladium length	(PL)	cm	Shape of the margin between areolas	(SMA)
Distance between areolas	(DBA)	cm	Areola coloring	(AC)
Leaf rib width	(LRW)	cm	Thorn color	(TC)
Height of undulations between successive areoles in a rib	(HUA)	cm	Pigmentation at the tips and margins of vegetative shoots	(PTM)
Number of spines per areola	(NSA)	#	Presence of wax	(PW)
Longest spine length	(LSL)	cm		
**Morphoagronomic and physicochemical descriptors of the fruits**				
Fruit length	(FL)	mm	Fruit shape	(FS)
Fruit width	(FW)	mm		
Fruit length/width ratio	(FRLW)	cm		
Fruit weight	(FWE)	g		
Pericarp thickness	(PT)	g		
Shell weight	(SW)	g		
Pulp weight	(PWF)	g		
Peel/pulp ratio	(PPR)	#		
Soluble solids	(SS)	(° brix)		
Acidity	(A)	%		
Number of bracts	(NB)	#		
Equatorial bract length	(EBL)	cm		
Equatorial bract width	(EBE)	cm		
Bract Color Index	(BCI)			

Symbol # refers to quantity.

## Data Availability

The data generated within this work are open access and available to be shared with interested persons.

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
