# Peer review of "Phenotypic Diversity of Morphological Characteristics of Pitahaya (Selenicereus Megalanthus Haw.) Germplasm in Colombia"

_plants, 2021, doi:10.3390/plants10112255_

Round 1
Reviewer 1 Report
Minor grammatical and language issues must be addressed before the manuscript may be published.
Author Response
The reviewer is thanked for his excellent recommendations, all of which were applied to the document.
The English of the entire manuscript was revised.
The suggested changes were made in each of the items in the manuscript.
The introduction and analysis of the results was supplemented with new bibliographic citations.
On behalf of all the authors, I appreciate all the comments made by the evaluator that really contribute to substantially improve the quality of the document.
Cordially,
ANA CRUZ MORILLO CORONADO
Author of correspondence

Reviewer 2 Report
The authors have significantly improved the manuscript. Therefore it can be accepted for publication after a very careful English language and Grammer check.
Author Response
The reviewer is thanked for all comments made on the document.
- The introduction of the work was corrected to better support the general objective of the research. Germplasm information is included and it is revealed that the research problem is that there is no certified planting material, therefore, in the fields different types of materials can be observed and with agronomic management according to tradition or experience in cultivation. Previous characterization studies are included at the international and national level, from which the descriptors used in this study are selected.
- On the other hand, the importance of hybridization is mentioned as a strategy to increase variability, especially in an autogamous species with preferentially asexual reproduction.
- In Colombia, there is conserved germplasm which has preferably been formed from collections carried out in the department of Boyacá, especially in the province of Lengupá, for which these materials were included in this study.
- In the materials and methods section, the municipalities in which the study is carried out are mentioned and their geographical location is shown in Table 1.
- Regarding the selected genotypes, in each of the productive systems visited, a molecular study with ISSR markers (unpublished data) was also taken into account to avoid redundancies and to be analyzing the same genotypes.
- The statistical analysis section was rewritten, delving into those aspects suggested by all the evaluators.
- Figures were included to help improve understanding of the article.
- A general revision of English was made throughout the document.
- All changes can be verified in the document.
On behalf of all the authors, I am grateful for all the contributions made by the reviewer, which contributed to improving the quality and understanding of all the research carried out.
Cordially,
Ana Cruz Morillo Coronado
Corresponding author

This manuscript is a resubmission of an earlier submission. The following is a list of the peer review reports and author responses from that submission.
Round 1
Reviewer 1 Report
This study evaluated the genetic diversity of yellow dragon fruit materials from different municipalities in the department of Boyacá with morphological descriptors to understanding the genetic background of this germplasm. The authors used three kinds of different clustering analyses, according to different data types, to investigate variation of phenotypic data. However, the findings cannot really be recognized as new ones. Literatures on diversity investigation and results are missing. I would suggest to focus it more on diversity analyses and on the findings to link with phenotypic diversity of the germplasm.
Introduction:
- In the introduction literatures on diversity and core collection of dragon fruit are missing as well as literatures about the introduced algorithms applied to problems like the mentioned ones. Problems and research approach and aim are not formulated as clear as possible.
- For this research topic about yellow dragon fruit, I think, it is worth to mention this germplasm material as well as the approaches for investigating phenotypic diversity.
Materials and methods:
- How many main producing municipalities in the department of Boyaca? The authors mentioned that “…in total five municipalities and 22 farms were sampled” on lines 345-346. However, there are only four municipalities listed in Table 1. How the 5 municipalities and 22 farms were sampled? What kind of sampling scheme was applied to the municipalities as well as to the farms? The authors have to describe the rationales of their sampling approaches.
- There is no information for materials (i.e. yellow dragon fruit germplasm). The selection of experimental materials and the description of the source of phenotypic data are not clear and detailed enough. How many germplasm accessions were included in this study?
- Did they process any data quality control analyses for phenotypic data before statistical analyses? Did they examine redundant accessions among their germplasm accessions?
- There were some missing phenotypic data. How did the authors handle the missing data? Since many multivariate statistical methods (like cluster analysis) can be done only for complete data (that is no missing data). Did you delete the accessions with missing data and use the remaining accessions for clustering analysis?
- Please clearly describe how the phenotypic data was obtained. Whether they were obtained under the same environment or not?
- How were genotypes of yellow dragon fruits selected?
- Sampling scheme for plants and fruits were not clear. The authors have to specify more carefully.
- The reason for correlation analysis is not clear. The authors should also specify it.
- The Euclidean distance is used for quantitative data. The authors need to describe more carefully how the Euclidean distance was applied for the cluster analysis?
- Also, there is no information for cluster analysis for mixed type data.
- The methods for the FactoMine R package is not clear. How were the Euclidean distance and hierarchical grouping method of Ward’s minimum variance applied to the mixed type data? The authors need to make it more clear.
- Shannon-Weaver diversity index and Nei’s diversity index (Shannon and Weaver, 1962; Nei, 1973) are popular methods for diversity investigation. I strongly suggest the authors to compute both Shannon-Weaver diversity index and Nei’s diversity index for the germplasm accessions.
Results:
- The authors should describe data analysis for climatic conditions in more details. What is this study duration? What type (daily/monthly data?) of these climatic data? In addition, the authors have to examine whether existing any significant differences among climatic data.
- The variables that make the greatest contribution to the variation of CP1 (also CP2) were not clear enough. Some important variables (e.g. FL and EBL for CP1, HUA and DBA for CP2) were not described.
- Did the cluster analysis use complete data (i.e. all selected germplasm accessions) or part of accessions (i.e. some accessions were deleted due to missing phenotypes)?
- How to determine five clusters in your cluster analysis? What criteria were used for groups selection?
- The authors mentioned “These analyzes were consistent with the principal component analysis.” on line 157. I checked Figure 3a and 3b. I really cannot follow it. Please explain it.
- Figure 4b was not matched to the results in lines 186-199. Figure 3b and 4b are the same.
Discussion:
- Can the authors really estimate one special value of one special accession from the other ones. Then they, for my opinion, must be related somehow? And that for diverse landraces or genotypes from different species? Perhaps you can clear that a little bit.
- The authors used 3 different methods for cluster analysis according to different data types. The results for the three cluster analyses were not consistent. It should be more reasonable to conduct cluster analysis for mix-type data. They should discuss this point more carefully.
- There is no limitations for this study.
Tables and Figures:
- The title of tables/figures were not written in English.
- There are no table and figure legends. It is difficult to read tables and figures.
- How many accessions were analyzed in Table 3?
Reviewer 2 Report
The objective of the paper is to characterize morphologically different genotypes from Selenicereus megalanthus collected in Colombia to provide grounds for selecting the best plants in dragon fruit plantations.
I find that the objectives of the paper fall short because authors only considered morphological characters without taking into account the genetic variation of cultivars as well as the chemical properties of fruits. Moreover they name genotypes without previous data that corroborates whether indeed they have different origin. Information on distribution of the species is lacking as well as differences production of fruits in different areas and its importance in the fruit market of Colombia. Also, justification on selection of characters is not discussed.
In Introduction previous research on different aspects of this species of dragon fruit is not mentioned. For example it has been stated the hybrid origin of Selenicereus megalanthus which is the species with the highest number of chromosomes in the genus. This is crucial to propose selection of cultivars for this study. Also, there are several collections in the wild, at least two in Colombia and Argentina and in my opinion they should be considered for analyses. This species has been reported in Brazil too. In addition, collections only come from a region, thus authors need to state whether the objective is restricted to a region in Colombia.
It is known as well that there is not genetic diversity in general in cacti and in particular for the Hylocereeae where this species belongs. Thus, without precise information authors can be analyzing the same genotypes.
Analyses are appropiate, however given the flaws mentioned above on sampling results are for a region in Colombia. Maybe they were collected where production is the highest in Colombia, however this was not mentioned. Furthermore justification for the number of samples should be provided as well. Graphs are well presented, and I suggest images of the variation of the fruit and the vegetative characters, and a more detailed map with the localities.
With regard to the English of the paper, I suggest authors use a specialized service and translate to English all the paper, some of the headings like the one from Table 3 is in Spanish and I found a number of words in Spanish (e.g. origen).
Thus I suggest these substantial changes for the paper
Reviewer 3 Report
My comments for each section are as below:
Abstract and introduction:
- The abstract must include data regarding the critical finds by the authors in terms of data of important findings.
- The introduction must have a clear hypothesis and significantly develop the second paragraph of this manuscript.
Results: Be careful of English usage and data values corresponding to the write-up in the text.Check figure ligands; they are carelessly written.
Discussion:
Discussion should include more information and references related to the relevant and related works.
Check the English in the Discussion section.